# Human and Mouse TRPA1 Are Heat and Cold Sensors Differentially Tuned by Voltage

**DOI:** 10.3390/cells9010057

**Published:** 2019-12-24

**Authors:** Viktor Sinica, Lucie Zimova, Kristyna Barvikova, Lucie Macikova, Ivan Barvik, Viktorie Vlachova

**Affiliations:** 1Department of Cellular Neurophysiology, Institute of Physiology of the Czech Academy of Sciences, 142 20 Prague, Czech Republic; viktor.synytsya@fgu.cas.cz (V.S.); tynabarvikova@seznam.cz (K.B.); lucie.macikova@fgu.cas.cz (L.M.); 2Department of Physical and Macromolecular Chemistry, Faculty of Science, Charles University, 128 00 Prague, Czech Republic; 3Division of Biomolecular Physics, Institute of Physics, Faculty of Mathematics and Physics, Charles University, 121 16 Prague, Czech Republic; ibarvik@karlov.mff.cuni.cz

**Keywords:** TRP channel, thermoTRP, noxious heat, noxious cold, transient receptor potential, ankyrin receptor subtype 1

## Abstract

Transient receptor potential ankyrin 1 channel (TRPA1) serves as a key sensor for reactive electrophilic compounds across all species. Its sensitivity to temperature, however, differs among species, a variability that has been attributed to an evolutionary divergence. Mouse TRPA1 was implicated in noxious cold detection but was later also identified as one of the prime noxious heat sensors. Moreover, human TRPA1, originally considered to be temperature-insensitive, turned out to act as an intrinsic bidirectional thermosensor that is capable of sensing both cold and heat. Using electrophysiology and modeling, we compare the properties of human and mouse TRPA1, and we demonstrate that both orthologues are activated by heat, and their kinetically distinct components of voltage-dependent gating are differentially modulated by heat and cold. Furthermore, we show that both orthologues can be strongly activated by cold after the concurrent application of voltage and heat. We propose an allosteric mechanism that could account for the variability in TRPA1 temperature responsiveness.

## 1. Introduction

The transient receptor potential (TRP) channel subtype A1 (TRPA1) is a polymodal sensor that is implicated in thermal and chemical nociception. Across different species, this channel particularly serves as a key receptor for electrophilic irritant compounds, evoking defensive responses [1,2,3]. The temperature sensitivity of TRPA1, on the other hand, has been reported to be a less evolutionarily conserved activation mode or alternatively occurring at the expense of chemical sensitivity [2,4]. The physiological role of this channel as a heat sensor has likely changed during evolution, potentially altering the preferred temperature ranges among species [5,6] or enabling some of them to detect and transduce infrared signals [4]. In invertebrates and ancestral vertebrates, TRPA1 acts as a heat-sensing ion channel [4,7,8,9,10,11,12]. In mammals, TRPA1 has long been considered to function as a cold receptor [13,14,15,16,17,18]. More recently, however, it has been also found to mediate a crucial physiological role in the detection of noxious heat [19,20,21]. A direct cold activation of TRPA1 has been demonstrated in several laboratories for mouse, rat and human orthologues [13,14,17,22,23], but it has not been observed by some other groups [4,24,25,26,27]. A substitution of a single specific amino acid residue (S250N) within the N-terminus of mouse TRPA1 results in a warm-sensitive channel [28], whereas human TRPA1 can become heat-activated when its N-terminal domain is transplanted from a rattlesnake orthologue [4]. Moreover, human TRPA1 reconstituted in an artificial membrane can act as an intrinsic bidirectional thermosensor that is capable of sensing both cold and heat [29], underscoring that the temperature-dependent behavior of the channel can be more complex than previously anticipated. A thorough perusal of all the available literature has indicated that there is still no common understanding of whether and, if so, under which conditions TRPA1 can directly act as a temperature sensor in mammals.

The sensitivity of TRPA1 to cold differs between primate and rodent species [26], and a single residue within the fifth transmembrane domain S5 has been proposed to underlie the observed differences. By using a chimeric domain swap analysis between rat and human TRPA1, the authors identified V875 in primates, corresponding to G878 in rodents, and demonstrated that the G878V mutation abolishes the cold activation of the rat and mouse TRPA1. Importantly, the same S5 region is involved in the sensitivity of TRPA1 to several nonelectrophilic agonists such as menthol, carvacrol, thymol, eudesmol, and protons, but also some pungent general anesthetics such as isoflurane, desflurane and propofol [30,31,32,33]. Moreover, mutations in S5 can strongly affect voltage-dependent characteristics of the channel [34] and the residues adjacent to V875 (S873 and T874) form, together with S6 (F944 and V948) and the first pore helix (F909), a binding pocket for the potent inhibitor A-967079 [31]. Most likely, the region around V875 is essentially involved in gating, and the sensitivity of TRPA1 to cold may thus depend on a proper allosteric coupling of a putative temperature-sensitive module to the gate domain.

The activation of certain temperature-sensitive TRP channels is associated with a temperature-induced a shift of their current–voltage relationships, and a simple two-state model has been proposed for a formal description of experimental measurements of channel activity [35]. Though this model seems to appropriately fit the voltage- and cold-dependent currents from mouse TRPA1 under some conditions [17], alternative modular allosteric models have been shown to be well-capable of capturing the characteristic the allosteric nature of thermo-TRP channel gating [36,37,38,39,40]. These models assume that the temperature and voltage sensors are separable parts of the protein that are allosterically coupled to the gate domain and to each other, and the transitions between the closed and open states are stimulus-independent. Particularly, the inverted coupling hypothesis was recently proposed by Jara-Oseguera and Islas [39], who demonstrated that the activation of cold-sensitive channels could be determined by the nature of the allosteric coupling between the heat-activated temperature sensor and the gate. We reasoned that TRPA1 is a prime candidate for testing such a predicted mechanism because some of the above studies have strongly suggested the modular nature of the temperature-dependent gating. Hence, we set out to compare the temperature- and voltage-dependent properties of human and mouse TRPA1 with the aim to decipher the fundamental differences between their mechanisms of gating.

## 2. Materials and Methods

### 2.1. Cell Culture, Constructs and Transfection

Human embryonic kidney 293T (HEK293T; ATCC, Manassas, VA, USA) cells were cultured in Opti-MEM I media (Invitrogen, Carlsbad, CA, USA) supplemented with 5% fetal bovine serum. The magnet-assisted transfection (IBA GmbH, Gottingen, Germany) technique was used to transiently co-transfect the cells in a 15.6 mm well on a 24-well plate coated with poly-L-lysine and collagen (Sigma-Aldrich, Prague, Czech Republic) with 200 ng of green fluorescent protein (GFP) plasmid (TaKaRa, Shiga, Japan) and with 400 ng of cDNA plasmid-encoding, wild-type or mutant human TRPA1 (pCMV6-XL4 vector, OriGene Technologies, Rockville, MD, USA) or mouse TRPA1 in pcDNA5/FRT vector, kindly provided by Dr. Ardem Patapoutian (Scripps Research Institute, La Jolla, CA, USA). The cells were used 24–48 h after transfection. At least three independent transfections were used for each experimental group. The wild-type channel was regularly tested in the same batch as the mutants. The mutants were generated by PCR by using a QuikChange II XL Site-Directed Mutagenesis Kit (Agilent Technologies, Santa Clara, CA, USA), and they were confirmed by DNA sequencing (Eurofins Genomics, Ebersberg, Germany). F11 cells (The European Collection of Authenticated Cell Cultures, ECACC 08062601, Porton Down, UK) cultured in Dulbecco’s Modified Eagle’s Medium supplemented with 2 mM glutamine and 10% fetal bovine serum were passaged once a week by using Trypsin-EDTA (Invitrogen, CA, USA) and grown under 5% CO_2_ at 37 °C. One-to-two days before transfection, cells were plated in 24-well plates (2 × 10^5^ cells per well) in 0.5 ml of medium, and they became confluent on the day of transfection. The cells were transiently co-transfected with 400 ng of cDNA plasmid-encoding mouse (in the pcDNA5/FRT vector) or human (in the pCMV6-XL4 vector) TRPA1 and with 200 ng of the GFP plasmid (TaKaRa, Japan) with the use of Lipofectamine 2000 (Invitrogen, CA, USA) and then plated on poly-l-lysine-coated glass coverslips.

### 2.2. Electrophysiology

Whole-cell membrane currents were filtered at 2 kHz by using the low-pass Bessel filter of the Axopatch 200B amplifier, and they were digitized (5–10 kHz) with a Digidata 1440 unit and the pCLAMP 10 software (Molecular Devices, San Jose, CA, USA). Patch electrodes were pulled from borosilicate glass and heat-polished to a final resistance of between 3 and 5 MΩ. Series resistance was compensated by at least 60%. The experiments were performed at room temperature (23–25 °C). Only one recording was performed on any one coverslip of cells to ensure that recordings were made from cells that had not been previously exposed to temperature or chemical stimuli. Extracellular bath solutions were Ca^2+^-free and contained: 140 mM NaCl, 5 mM KCl, 2 mM MgCl_2_, 5 mM EGTA (ethylene glycol-bis (β-aminoethyl ether)-*N*,*N*,*N*′,*N*′-tetraacetic acid), 10 mM 4-(2-Hydroxyethyl)piperazine-1-ethanesulfonic acid (HEPES), 10 mM glucose, and pH 7.4 was adjusted by tetramethylammonium hydroxide. The intracellular solution contained 140 mM KCl, 5 mM EGTA, 2 mM MgCl_2_, 10 mM HEPES, adjusted with KOH to pH 7.4. For F11 cell recordings, the internal pipette solution contained 125 mM Cs-gluconate, 15 mM CsCl, 5 mM EGTA, 10 mM HEPES, 0.5 mM CaCl_2_, 2 mM MgATP, 0.3 NaGTP, and 10 mM HEPES, adjusted to pH 7.4 with CsOH, 283 mOsm. All of the chemicals were purchased from Sigma-Aldrich (Prague, Czech Republic). The *I–V* relationships were recorded by using 400-ms steps ranging from −150 to +100 mV, followed by a 400-ms step to −150 mV (increment +25 mV) and a holding potential of 0 mV, or they were recorded by using 100-ms steps from −80 mV to +200 mV (increment +20 mV) and holding potential of −70 mV.

### 2.3. Temperature Stimulation

A system for the fast cooling and heating of solutions that superfuse isolated cells under patch-clamp conditions was used as described previously [41]. Briefly, experimental solutions are driven by gravity from 7 different barrels through automatically controlled valves to a manifold that consists of fused silica tubes (320 µm inner diameter) that are connected to a common outlet glass capillary through which the solutions are applied onto the cell surface. The upper part of the outlet capillary passes the solutions through a temperature exchanger driven by a miniature Peltier device that preconditions the temperature (precooling or preheating). The lower part of the capillary is wrapped with an insulated, densely-coiled copper wire (over a length of 5 mm, connected to a direct current source for resistive heating) that finally heats the passing solution to a chosen temperature. The Peltier device and the heating element are electrically connected to the headstage probe, which is fixed on to a micromanipulator for the positioning of the manifold. The temperature of the flowing solution is measured by a miniature thermocouple that is inserted into the common outlet capillary near to its orifice that is placed at a distance of less than 100 µm from the surface of the examined cell. The temperature is controlled either automatically or manually by voltage commands via the digital-to-analogue converter of a conventional data acquisition interface (Digidata 1440, Molecular Devices). The system allows for the application of temperature changes within a range of 5–60 °C at maximum rates between −40 and ~110 °C/s.

### 2.4. Molecular Modelling

To elucidate the possible structural mechanism by which the substitutions at S804 (S804D, S804N, S804A) produce dramatic functional changes, we used the human TRPA1 structure with modeled S1–S2 and S2–S3 linkers, which is available in the Model Archive (www.modelarchive.org) under the accession code ma-auqu1 [42] and which is based on the structure with Protein Data Bank (PDB) ID: 3J9P, determined by cryo-electron microscopy [31]. The TRPA1 tetrameric structure was inserted into the patch of the 1-palmitoyl-2-oleoylphosphatidylcholine (POPC) bilayer and solvated in transferable intermolecular potential 3-point (TIP3P) [43] water molecules to ensure the presence of at least 10 Å of solvent on both sides of the membrane, and then the structure was neutralized in 0.5 M NaCl. This gave a periodic box size of 120 × 120 × 160 Å for a simulated system consisting of ~205,000 atoms. All atom structure and topology files were generated by using VMD software [44]. Forces were computed by using the CHARMM27 force field for proteins, lipids, and ions [45,46,47]. All molecular dynamics (MD) simulations were produced with the aid of the software package NAMD2.13 [48] running on computers equipped with NVIDIA graphics processing units (Nvidia Corporation, Santa Clara, CA, USA). First, Langevin dynamics were used for temperature control, with the target temperature set to 285 K, the Langevin piston method was applied to reach an efficient pressure control with a target pressure of 1 atm [48]. The integration timestep was set to 2 fs. Simulated systems were energy-minimized, heated to 285 K, and production MD runs reached a length of 2000 ps. Data were recorded every 2 ps, and contacts of S804 (or S804D, S804N) with ambient polar residues within 12 Å were analyzed by using the CPPTRAJ module from Amber Tools suite [49].

To obtain a model of human TRPA1 with the C-terminal loop region (1007–1030), a homology model of the C-terminal loop region was created by using the Swiss-Model web server (https:/swissmodel.expasy.org/) and incorporated into the 3J9P structure. Then, we used the original electron density map of human TRPA1 with the Electron Microscopy Data Bank (EMDB) ID: 6267 [31] and applied the molecular dynamics flexible fitting (MDFF) method for combining high-resolution structures with cryo-electron microscopy (cryo-EM) maps, a method that creates atomic models that represent the conformational state captured by cryo-EM [50,51]. We followed standard procedure by using software packages, modules, and keywords described in detail in “MDFF Tutorial” (http://www.ks.uiuc.edu/Training/Tutorials/). MD trajectories were visualized with the aid of the VMD 1.9 software package [44]. Figures were produced with the software packages UCSF Chimera 1.13 [52] and CorelDraw X7 (Corel Corporation).

### 2.5. Statistical Analysis

The electrophysiological data were analyzed by using pCLAMP 10 (Molecular Devices), and the curve fitting of currents and statistical analyses were done in SigmaPlot 10 (Systat Software Inc.). Conductance–voltage (*G–V*) relationships were obtained from the steady-state whole cell currents measured at the end of voltage steps. Voltage-dependent gating parameters were estimated by fitting the conductance *G* = *I*/(*V* − *V*_rev_) as a function of the test potential *V* to the Boltzmann equation: *G* = [(*G*_max_ − *G*_min_)/(1 + exp (−*zF*(*V* − *V*_50_)/*RT*))] + *G*_min_,(1)
where *z* is the apparent number of gating charges, *V*_50_ is the half-activation voltage, *G*_min_ and *G*_max_ are the minimum and maximum of the whole-cell conductance, *V*_rev_ is the reversal potential, and *F*, *R*, and *T* have their usual thermodynamic meanings. Activation time constants (τ_fast_ and τ_slow_) were directly measured from outward currents by performing a double exponential Chebyshev fit to the rising phase of the activating currents. Fast and slow deactivation was characterized by a single or a double exponential fit to the current upon repolarization. The initial 1.3 ms interval was ignored. For comparisons between human TRPA1 (hTRPA1) and mouse TRPA1 (mTRPA1), we calculated the average weighted time constants of the activation and deactivation process based on the amplitudes of the fast and slow components: τ_w_ = *A*_fast_/(*A*_fast_ + *A*_slow_) × τ_fast_ + *A*_slow_/(*A*_fast_ + *A*_slow_) × τ_slow_,(2)

Statistical significance was determined by Student’s *t*-test or the analysis of variance, as appropriate; differences were considered significant at *p* < 0.05 where not stated otherwise. The data are presented as means ± (or +/−) SEM.

### 2.6. Kinetic Modeling and Simulations

The open probability versus voltage and temperature, *P*_o_ (*V*,*T*), landscapes were generated by using the analytic expression for the open probability (Equation (4) in [39]), written as a data transformation script in SigmaPlot 10 (Systat Software Inc., San Jose, CA, USA). The steady-state data were fit to an eight-state allosteric model that was written in COPASI 4.22 [53] by using the built-in genetic algorithm. The equilibrium constant of the temperature sensor *J*(*T*) was set to the value published for TRPM8 [36,39]: Δ*H*^o^ = 91 kcal mol^−1^ and Δ*S*^o^ = 0.317 kcal mol^−1^ K^−1^. The parameter Δ*H*^o^ was kept constant for all modeled situations. The parameters *K*(0), *z*, *L*, *E*, *D*, Δ*H*^o^*_C_*, and Δ*S*^o^*_C_*were estimated by fitting the average conductance to voltage (*G*/*V*) curves obtained for hTRPA1 at 25 and 12 °C. To qualitatively match the activation profile of a representative hTRPA1 recordings, two additional *P*_o_ (*V*,*T*) landscapes were calculated. The “green” and “red” *P*_o_ (*V*,*T*) landscapes (shown in the Results section below) were modeled to meet the assumptions coming from the activation profile of hTRPA1, normalized to *G*_max_. *P*_o_ = 0.15 at −70 mV and 60 °C, *P*_o_ = 0.005 at −70 mV and 5 °C, *P*_o_ = 0.21 at +80 mV and 5 °C, and *P*_o_ = 0.71 at +80 mV and 60 °C (for green); *p* = 0.91 at −70 mV and 60 °C, *P*_o_ = 0.998 at −70 mV and 5 °C, *P*_o_ = 0.23 at +80 mV and 5 °C, and *P*_o_ = 0.74 at +80 mV and 60 °C (for red). To obtain the green *P*_o_ (*V*,*T*) landscape, all parameters from the blue *P*(*V*,*T*) landscape were fixed except for the allosteric coupling factors *E, D, C*(Δ*H*^o^*_C_*, Δ*S*^o^*_C_*) and equilibrium constant *L*. The parameters for the red landscape were obtained in two steps. In the first step, the parameters from the green *P*_o_ (*V*, *T*) landscape were fixed except for *D, L*, Δ*H*^o^*_C_*and Δ*S*^o^*_C_*. The obtained values for *D* and *L* were used in the second step to find parameters: *E*, Δ*S*^o^, Δ*H*^o^*_C_* and Δ*S*^o^*_C_*. The resulting parameters were then used to simulate the time-dependent probability for each state of the model over the time interval of 10–50 s with a time step of 0.1 s. The exemplary data used for our kinetic simulations were obviously not sufficient to unambiguously constrain the large number of free model parameters. We performed a large number of different simulations with different initial sets of free parameters. In some cases, the fits successfully converged, but the obtained values were physically unrealistic. However, when the sort of constraints described above was applied, the fits converged to similar solutions when different initial parameter values were used for several repeated fittings.

## 3. Results

### 3.1. Temperature and Voltage Dependence of Human and Mouse TRPA1

Previous studies have shown that the cold sensitivity of mTRPA1 can be described by a two-state model in which temperature directly affects the equilibrium between the open and closed states of the channel, based on global changes in enthalpy and entropy during channel gating [17]. The authors demonstrated that upon cooling, the voltage dependence of mTRPA1 channel activation is shifted towards more negative voltages, similarly to the behavior of certain other temperature-activated TRP channels [35]. To compare the voltage-dependence and relaxation kinetics between hTRPA1 and mTRPA1, we measured whole-cell currents in response to a voltage step protocol akin to that used previously by [17], consisting of 400-ms steps ranging from −150 to +100 mV, followed by a 400-ms step to −150 mV (Figure 1A–D). The currents were measured at 12, 25, and 35 °C under Ca^2+^-free conditions to prevent the Ca^2+^-dependent potentiation and inactivation of TRPA1 [54]. We constructed the conductance-to-voltage (*G*/*V*) relationships and found that cold (12 °C) and heat (35 °C) increased the conductance at negative membrane potentials in both channel orthologues (Figure 1B,D,E). An obvious difference was in a constitutive voltage independent conductance evident at 25 °C and negative membrane potentials in mTRPA1 (2.5 ± 0.7 nS at −150 mV; *n* = 23) but not in hTRPA1 (0.8 ± 0.1 nS; *n* = 7; *p* < 0.001). On the other hand, the maximum conductance at +100 mV measured in mTRPA1 at 25 °C (23.4 ± 10.9 nS) was not statistically different from that in hTRPA1 (23.5 ± 1.8 nS; *p* = 0.994), suggesting a shift in the basal gating equilibrium towards the open state at negative potentials rather than differences between the expression levels of the two channels. The half-maximum activation voltage (*V*_50_) of hTRPA1 was shifted from 66.5 ± 6.7 mV at 25 °C to 53.4 ± 3.0 mV at 35 °C (*p* = 0.035). In contrast, the voltage activation curve was not shifted upon warming in mTRPA1 (*V*_50_ of 60.9 ± 6.4 mV at 25 °C and 62.8 ± 6.3 mV at 35 °C; *n* = 6; *p* = 0.732). In both channels, warming from 25 to 35 °C did not significantly alter the apparent gating charges (hTRPA1: 0.76 ± 0.03 e_0_ and 0.85 ± 0.06 e_0_; *p* = 0.256; paired *t*-test; *n* = 8; mTRPA1: 0.85 ± 0.05 e_0_ and 0.82 ± 0.03 e_0_; *p* = 0.523; paired *t*-test; *n* = 6). At 12 °C, the voltage-induced currents did not reach steady state at the end of the step pulse, and the *G*/*V* relationships were estimated from the exponential fit of the recorded activation curves (Figure 1A,C).

To compare the effects of temperature on the activation and deactivation kinetics, we determined the time constants by mono- or bi-exponential fits to the time course of the onset currents at +100 mV and the inward tail currents measured at −150 mV (Figure 1E,F). The activation phase of the outward currents mediated by mTRPA1 at 25 °C could be fitted by a monoexponential function with a time constant, *τ*_on_, of 33.0 ± 2.7 ms (*n* = 10). In contrast, an adequate fit of the activation phase of the currents through hTRPA1 required at least two exponential functions with the average fast and slow components of 39.3 ± 3.3 ms (contributing by 80.6 ± 5.6%) and 155.4 ± 24.7 ms (*n* = 25), respectively. The temperature dependencies of the activation rates were compared by plotting the inverse of *τ*_on_ versus 1/*T* (Figure 1F). The relationship for mTRPA1 could be fitted well by linear regression with a resultant slope of 87.6 kJ mol^−1^. Provided that the activation of mTRPA1 was limited by one main step, this value corresponded to *Q*_10_ = 3.3 within the 10–35 °C temperature range. In hTRPA1, the temperature dependencies of the fast and slow component of the activation process exhibited slopes of 95.1 kJ mol^−1^ (*Q*_10_ = 3.7) and 77.9 kJ mol^−1^ (*Q*_10_ = 2.9)

The time course of the inward tail currents at −150 mV (Figure 1F, right) was monoexponential in all cells expressing mTRPA1 (*τ*_off_ at 25 °C, 9.1 ± 0.6 ms; *n* = 10), which is in agreement with previously published results [17]. In a clear contrast, the hTRPA1 tail currents could be well-fitted only by two exponentials, *τ*_fast_ of 5.0 ± 0.4 ms (contributing by 70.4 ± 5.2%) and *τ*_slow_ of 28.4 ± 1.8 ms at 25 °C (*n* = 23). The initial rapid component of hTRPA1 deactivation was fast at 35 °C, which made the determination of *τ*_fast_ uncertain. To reliably extrapolate the temperature dependency of the deactivation rate for the human orthologue, we additionally measured the voltage-induced currents at 20 and 30 °C. From the Arrhenius plot of the deactivation rates, it was apparent that the deactivation process of mTRPA1 exhibited a slightly steeper temperature dependence (81.3 kJ mol^−1^; *Q*_10_ = 3.1) than the fast component of the deactivation of hTRPA1 (67.2 kJ mol^−1^; *Q*_10_ = 2.5). These results indicate that the voltage-dependent activation and deactivation processes of both orthologues exhibited only mild temperature dependencies over the temperature range of 12–35 °C. Compared to hTRPA1, the gating equilibrium of mTRPA1 was significantly shifted towards the open state.

### 3.2. Mutations hTRPA-V875G and mTRPA1-G878V in S5

Previous studies have identified G878 in mTRPA1 and V875 in hTRPA1 (Figure 2A) as residues underlying the species-specific differences in cold sensitivity [26]. We addressed the question of to what extent this residue could contribute to the temperature dependence of the voltage-induced gating. We measured the voltage-activated currents from hTRPA-V875G and mTRPA1-G878V at 12, 25, and 35 °C (Figure 2B–E and Appendix A) and compared the temperature dependencies of the deactivation rate by fitting the time course of the inward tail currents at −150 mV by mono- or bi-exponential functions (Figure 2F–I). Most interestingly, the temperature dependence of the fast component of the deactivation rate of hTRPA1-V875G became steeper, thus resembling that of mTRPA1 (83.6 kJ mol^−1^; Figure 2G). The reverse mutation mTRPA1-G878V had an opposite effect: A decreased slope of the temperature dependence of the deactivation rate approached the less steep slope of the wild-type hTRPA1 (69.6 kJ mol^−1^; Figure 2I). These results not only complement those obtained by Chen et al. [26] but also further emphasize the possible role of the residue V875/G878 in S5 in allosteric gating.

### 3.3. Functional Role of Non-Homologous Residues in the Vicinity of V875

Next, we wondered how the temperature signal might be conveyed to the gate and how the allosteric coupling or gating of primate and rodent orthologues is adjusted by the vicinity of V875/G878. By mapping all residues that are non-conserved between mouse and human TRPA1 onto the cryo-EM structure, we identified several sites in the inner part of the S1–S4 sensor and the S4–S5 linker located close to S5 (Appendix A). Accordingly, we constructed and functionally analyzed a set of reverse mutants of hTRPA1 (M801L, S804N, V806A, and I803Y/L867F) at 25, 12, and 35 °C. Among them, the single human to mouse mutation S804N was most affected, as it exhibited large basal currents, very slow activation and deactivation kinetics, and a strong voltage-independent component (Figure 3A–C and Appendix A). The half-maximum activation voltage, *V*_50_, was drastically shifted to 21.6 ± 4.1 mV at 25 °C and to 29.5 ± 4.6 mV at 12 °C (*n* = 7; *p* < 0.001). The steady-state conductance measured at 35 °C almost overlapped with the *G*/*V* relationships measured at room temperature (Figure 3B). From the Arrhenius plot of the deactivation rates, it was apparent that the deactivation process of S804N exhibited a somewhat steeper temperature dependence (103.8 kJ mol^−1^; *Q*_10_ = 4.2) compared to the wild-type channels (Figure 3C). To better understand the biophysical requirements at position 804, we explored small nonpolar and charged substitutions. The replacement of the serine with an alanine, S804A, rendered the channel nonfunctional (Figure 3D), whereas aspartate substitution resulted in a phenotype similar to S804N (Figure 3E) in that the activation of S804D by cold produced large and persisting currents at negative membrane potentials (see Appendix A below).

### 3.4. Molecular Modeling of the S804 Mutants

To explore the possible structural mechanism by which the substitutions at S804 produce such dramatic functional changes, we performed molecular simulations by using the structure of human TRPA1 (PDB code 3J9P) [31] completed with the S1–S2 and S2–S3 linkers [42,55]. The molecular dynamics (MD) run with the wild-type TRPA1 identified rather loose contacts between S804 and either R852 or N845 from S4 (Figure 3F,G). In contrast, the N804 side chain was involved in hydrogen bonding exclusively with N845, while the D804 side chain formed tight salt bridges with R852 in all monomers of the channel structure. The serine 804 was found to be a part of the lower cavity formed by the S1–S4 sensor domain, previously shown to be an important regulatory site of TRPA1, integrating the regions from the TRP-like domain, the S4–S5 linker, and S5 [42]. The results from our MD simulations thus reinforce the previous observations and highlight the key role of conserved polar residues comprising the putative lower crevice of the sensor domain in the integration of the activation signals and their transmission to the intracellular channel gate.

### 3.5. Modeling the Proximal C-Terminal Loop

V875 appears to be directly impacted by the proximal C-terminal loop (Y1006-Q1031) that follows the TRP-like helix and contains a short, predicted α-helix that is partly buried in the inner leaflet of the membrane (Appendix A). This loop is integrated with the pre-S1 region and, together with the S2–S3 linker, comprises an intracellular vestibule that forms a side ‘fenestration’ through which hydrophobic modulators and membrane phospholipids may affect channel gating [31]. Because the region Y1006-Q1031 is not well-resolved in the published cryo-EM structure [31], we used the molecular dynamics flexible fitting (MDFF) approach to refine fitting of the atomic structure to the density map and to gain an insight into the nature of the mechanical coupling within the channel. We found interactions between V875 and the subregion of the C-terminal loop L1019–F1022 with some variability in different TRPA1 monomers (Appendix A). However, outside the area of the density with the predicted α-helix, the conformational promiscuity of the fitted protein chains was much more pronounced. This indicates rather loose contacts between this C-terminal segment and the remaining transmembrane core of TRPA1, which may be strengthened by the binding of various hydrophobic modulators.

To verify our structural model, we constructed and analyzed three reverse mutants based on the comparisons between rodent and primate TRPA1 sequences (Appendix A): H1018R, ΔG1013 and G1027Q. The first residue was predicted to be located in close proximity to V875, whereas the latter two mutants seemed to have a good chance of reducing the conformational fluidity of the C-terminal region. We evaluated the temperature dependence of the steady-state conductance (Appendix A) and of the faster, dominant component of the deactivation rate measured from the ‘tail currents’ following the repolarization from +200 to −70 mV (Appendix A). The ΔG1013 and H1018R constructs had significantly (*p* ≤ 0.001) faster deactivation kinetics in common at 13 °C (58.3 ± 3.2 s^−1^ and 61.1 ± 9.1 s^−1^; *n* = 4 and 6) compared with wild-type hTRPA1 (38.6 ± 2.1 s^−1^; *n* = 20). Within the 13–24 °C temperature range, the latter mutant exhibited slightly less steep temperature dependence (61.1 kJ mol^−1^) than hTRPA1 (75.0 kJ mol^−1^). G1027Q was not significantly different from wild-type hTRPA1, indicating that these residues are not key determinants of temperature sensitivity.

Previous studies have identified the C-terminal loop as a site for Ca^2+^/calmodulin and phosphatidylinositol-4,5-bisphophate binding [54,56]. Mutations within this region affected TRPA1 responses induced by various stimuli including cinnamaldehyde, allyl isothiocyanate, voltage, Ca^2+^ and carvacrol. Thus, it seems most likely that the C-terminal loop serves as a universal regulatory domain that integrates signals from different parts of the protein, and its role in temperature sensitivity is not specific. This is further supported by the fact that introducing domain containing C-terminus of hTRPA1 did not affect cold activation of rodent TRPA1 [26].

### 3.6. Effects of Temperature on TRPA1 at a Constant Membrane Voltage

Our results indicate that hTRPA1 and mTRPA1 exhibit a dual warm and cold sensitivity that is apparent at negative membrane potentials (Figure 1B,D). This so-called U-shape thermosensitivity has been previously reported for purified hTRPA1 reconstituted into planar lipid bilayers [29], thus suggesting that the channel is inherently cold- and heat-sensitive. An allosteric eight-state model capable of replicating the U-shape thermosensitivity of thermal channels assumes that the channel possesses heat and voltage sensors allosterically coupled to the gate and to each other [39]. To explore the significance of the coupling energy between the voltage and temperature sensors, we measured currents at +80 mV by using increasing steps of temperature from 11 to ~50 °C in 3-second intervals (Figure 4A). Obviously, both TRPA1 orthologues exhibited U-shaped outward currents with a saddle point around the room temperature (Figure 4B). Though the hTRPA1-mediated currents were slightly shallower than those through mTRPA1, it was evident that the responses of the both channels were increased upon cooling and heating. Moreover, we reasoned that if the Arrhenius plot of the currents was extrapolated to even higher temperatures, TRPA1 could exhibit the properties of a heat-activated ion channel.

Therefore, we measured currents at −80 and +80 mV from hTRPA1 and mTRPA1 in a control bath solution by using temperature ramps from 25 to 60 °C that were applied at a maximum speed of about 35 °C/s in 1.5 s intervals (Figure 5A–C). Indeed, we observed specific heat-induced currents at negative and positive membrane potentials that were rapidly reversible and exhibited steep temperature dependence over the high temperature range of 53–59 °C for mTRPA1 and 55–57 °C for hTRPA1 (Appendix A). The heat-induced activation of mTRPA1 has been previously observed in sensory neurons but not in non-neuronal cell lines, and it has thus been attributed to an indirect activation by some unknown mechanism [21]. Our data show that hTRPA1 and mTRPA1 can be directly, rapidly, and reversibly activated by noxious heat in non-neuronal cells, indicating that the both mammalian orthologues possess an intrinsic heat-sensing domain. The heat-induced currents measured at positive holding potentials were substantially greater than those at negative voltages, suggesting an allosteric mechanism of activation. Therefore, we next explored to what extent the allosteric coupling between the voltage- and heat-sensors may account for the observed variability in temperature responses.

### 3.7. Interactions between Temperature- and Voltage-Sensor Modules

As evident from Figure 1A–F, the slower components of the activation and deactivation voltage-dependent characteristics of both TRPA1 orthologues could be substantially underestimated at cold temperatures over the time interval used for recording. To allow the channels to fully activate and relax back to equilibrium, we applied 10-s long depolarization pulses from −70 to +80 mV at a constant cold temperature (~5–10 °C) and tested the effects of exposure to noxious (~60 °C) heat. When the temperature was stepped for ~5 s below 60 °C, the subsequent response to +80 mV did not significantly differ from the preceding one. However, if the heat step exceeded 60 °C, the subsequent response to +80 mV was much smaller, and the degree of reduction was strongly correlated to the maximum temperature applied (Figure 6A–D; Pearson correlation; *p* = 0.0005 and *p* = 0.0003 for hTRPA1 and mTRPA1; *n* = 10 and 19). Exposure to excessive heat thus seems to irreversibly impede either the activation of the voltage-sensor or its effective coupling to the gate.

Most notably, when the channels were concurrently exposed to depolarization along with noxious heat above 60 °C, we observed strikingly large inward currents after cooling the cells to 5–8 °C and subsequent repolarization to −70 mV (Figure 7A–F). The cold-sensitized inward currents were produced independently of whether depolarization or heat was applied prior to the concurrent activation of both sensors (Figure 7E), but these currents were not seen after the heat- or voltage-sensors were individually activated/deactivated (Figure 7A,C,E,F). Similar cold-sensitized inward currents were observed in hTRPA1 and mTRPA1, as well as in the reverse mutants with slowed activation/deactivation kinetics (hTRPA1-S804D and mTRPA1-N807S; Appendix A).

The inverted coupling hypothesis proposed by Jara-Oseguera and Islas [39] states that activation of thermosensitive TRP channels by cooling could be mediated by a heat-activated temperature sensor that is energetically unfavorably coupled to the pore gating. Similarly, voltage-gated channels can be activated either by depolarization or hyperpolarization, depending on whether their voltage sensors are coupled favorably or unfavorably to the gate [57]. Our results indicated that noxious heat above 60 °C induces substantial structural rearrangements within the channel, leading to a strong inward rectification at cold temperatures. We therefore reasoned that the above hypothesis could be adopted to explain our observations, providing that either the heat- and/or voltage-sensors become unfavorably coupled to the gate as a consequence of the exposure to excessive heat.

### 3.8. Can the Temperature Sensitivity of TRPA1 Be Explained by Inverted Coupling Hypothesis?

We performed simulations by using the eight-state allosteric kinetic model that had previously been shown to be consistent with the gating of the TRPA1-related channels, TRPV1 and TRPM8 [36,39]. This model is based on the presumption that an independent voltage- and heat-sensors exist that are coupled to the channel gate and to each other (Figure 8A). At constant low temperatures and low voltages, the channel is confined to the equilibrium between closed and open states with an equilibrium constant *L*. The temperature dependence is characterized by the difference in enthalpy (*ΔH*^o^) and entropy (*ΔS*^o^) between the states. Temperature and voltage drive transitions between resting and activated states of the sensors with equilibrium constants *J*(*T*) and *K*(*V*), respectively. In turn, the temperature and voltage sensors are coupled to channel gating by allosteric coupling factors *C* and *D*, and the two sensors are coupled by the allosteric constant *E*. To involve the bidirectional thermosensitivity of the channels, the allosteric coupling factor *C* was assumed to be temperature-dependent: *C*(*T*) = exp[−(Δ*H*^o^*_C_* − *T*Δ*S*^o^*_C_*)/*RT*].

Using the parameters published for TRPM8 [39] as the initial values, we estimated *K*(0), *L*, *E*, *D, z*, Δ*H*^o^*_C_* and Δ*S*^o^*_C_* by fitting the average *G*/*V* curves obtained for hTRPA1 at 25 and 12 °C (Figure 8B). These parameters were then refitted to qualitatively match the activation profile of a representative hTRPA1 recording, such as that shown in Figure 7A, normalized to *G*_max_. We calculated the open probability (*P*_o_) as a function of voltage and temperature and plotted the *P*_o_(*V*,*T*) landscapes (Figure 8C–E). We found that the model was capable of capturing all of the essential aspects of our experimental observations (Figure 8F):
Initially, voltage-dependent activation was only mildly temperature-dependent over the temperature range of 12–35 °C, and cold only weakly activated the channels at negative membrane potentials.The channels were activated by high noxious temperatures, and the heat-induced responses were strongly outwardly rectifying (as shown in Figure 5).Exposure to excessive heat (>60 °C) irreversibly reduced subsequent responses to depolarizing potentials. This could be achieved by a strong (72.5-fold) increase in the coupling factor *E*, a 1.4-fold increase in the relative magnitude of the enthalpy and entropy values of the allosteric factor *C*, Δ*H*^o^*_C_*/Δ*S*^o^*_C_*, a 10-fold decrease in the gating equilibrium *L*, and a 1.07-fold decrease in the coupling factor *D* (Figure 8D).The channels became strongly inwardly rectifying after the concurrent activation of voltage and heat sensors and their subsequent deactivation. This could be simulated by an increase in *L* (271.9-fold) and *E* (5.27-fold), a decrease in the relative magnitude of the temperature sensor‘s enthalpy and entropy values (Δ*H*^o^/Δ*S*^o^, 2.1-fold), and the relative magnitude of the enthalpy and entropy values characterizing the allosteric factor *C* (Δ*H*^o^*_C_*/Δ*S*^o^*_C_*, 2.0-fold), and a large (32,733-fold) decrease in *D* (Figure 8E).The responses activated during heating typically exhibited current transients, suggesting that the channels passed through several conformational states before they reached their steady-state (Figure 8F,G). This further supports the modular allosteric model (rather than the two-state model) as an appropriate approximation of TRPA1 activation.

### 3.9. Activation of TRPA1 in F11 Cells

To investigate whether TRPA1 can be activated by heat and cold under close-to-native conditions, we used F11 neuroblastoma cells derived from dorsal root ganglia neurons, which provide a well-characterized cellular model of peripheral sensory neurons [58]. We transfected the cells with mouse or human TRPA1 and measured currents at −70 mV by using a similar temperature and voltage protocol to that shown in Figure 7A. In addition, the TRPA1 inhibitor HC-030031 (50 µM) was used to assess the extent of specific contribution of TRPA1 to temperature-dependent currents. We observed very similar current characteristics to those seen in channels expressed in HEK293T cells. Though the application of heat above 60 °C was experimentally challenging and inevitably destroyed the seal in many cells (19 of 29), we succeeded in recording the typical responses to heat and voltage in eight neurons that expressed mTRPA1 and two neurons expressing hTRPA1 (Appendix A). In both TRPA1 orthologues, we observed a reduction in currents induced by depolarizing step to +80 mV only when the preceding heat step exceeded 60 °C. The responses activated during heating typically exhibited current transients, suggesting that the channels passed through several conformational states, as in HEK293T cells (see Figure 8G). When the neurons were depolarized and concurrently stimulated by noxious heat, inward currents arising from cooling the cell to 5–7 °C and repolarizing it to −70 mV were increased (Appendix A) and slowed down (Appendix A).

In a large subset of native dorsal root ganglia neurons, TRPA1 physically and functionally interacts with the structurally-related, heat-sensitive vanilloid receptor subtype 1 channel TRPV1 [59]. Our results indicate that TRPA1 may respond to heat in isolation from TRPV1, which adds further complexity to the mechanisms that contribute to noxious heat transduction in somatosensory neurons.

## 4. Discussion

The data presented in this study demonstrate that both human and mouse TRPA1 orthologues have an intrinsic propensity to respond to excessive heat and adopt conformations primed for cold activation at physiological membrane potentials. Most importantly, our results unveil a specific mode of TRPA1 activation that can be viewed as a ‘heat-induced cold sensitivity.’ This transition occurs after the channels are activated simultaneously by depolarizing voltage and high noxious heat and then are subsequently repolarized to the resting membrane potential at cold temperatures.

Our experimental observations indicate that excessive heat renders the channels less activated by voltage. As predicted from the eight-state allosteric model, heat above 60 °C may affect the gating equilibrium constant—the extent of allosteric coupling of the voltage- and heat-sensors to the gate domain and to each other. When the two stimuli, heat and voltage, were applied together, the channels became strongly sensitized to cold. Our simulations predict that the concurrent activation of the voltage- and heat-sensors can induce a conformational switch that leads to an increase in their energetic crosstalk, an increase in the gating equilibrium constant, and a drastic (~30,000-fold) decrease in the coupling of the voltage sensor to the channel gate.

Substantial differences in the pharmacological profiles between rodent and primate orthologues seriously hamper current screening and medicinal chemistry discovery efforts for targeting TRPA1 [60,61]. Moreover, the distinct temperature sensitivities of these channels raise important questions about the physiological relevance of all the findings obtained in rodent models. Our findings that both TRPA1 orthologues exhibit qualitatively similar temperature- and voltage-dependent characteristics suggest a conserved molecular logic for the gating of TRPA1 across different species and may reconcile the seemingly conflicting lines of evidence in published literature. At negative membrane potentials (−150 mV) and at 25 °C, the gating equilibrium of mTRPA1 is shifted toward the open state (11.4% of *G*_max_) when compared with hTRPA1 (2.3%) (Figure 1B,D). From our predictions (using the parameters in Figure 8C while only changing *L*), this difference may lead to a 3–4-fold increase in responses induced by both heat (60 °C) and cold (5 °C) at −70 mV. Thus, the difference in the intrinsic gating equilibrium itself may account for the observed disparity between the two orthologues.

Moreover, hTRPA1 exhibits significantly faster deactivation kinetics that consist of two exponential components, one of these of a similar temperature dependence as mTRPA1 (Figure 1F). The slower voltage-dependent relaxation kinetics seen in mTRPA1 and in certain mutants of hTRPA1 (hTRPA1-S804D and S804N) allows these channels to dwell longer in the open state(s) at cold temperatures and, thus, produce significant and persistent currents at negative membrane potentials. This, together with the high sequence similarity (88.9% similarity and 79.8% identity), indicates that the difference between the two mammalian TRPA1 orthologues might lie not in their temperature- or voltage-sensing domains but rather in the mechanisms by which they are coupled to the channel gate and/or to each other. As TRPA1 acts as a heat-sensing ion channel in most species that were tested so far, the heat-sensitive domain could have been conserved during evolution while the evolutionary changes could serve to fine-tune its energetic coupling to the pore gate and/or the voltage-sensitive domain. From this point of view, it would be important to explore the temperature sensitivity of clones from various species under identical experimental conditions and, particularly, over as wide as possible a temperature range.

Since the first cloning and characterization in 2003 by Ardem Patapoutian’s group [13], the mouse TRPA1 has been considered to be cold-activated. Fifteen years later, its role in heat (45 °C) detection was also convincingly shown, although it was assumed that the heat responsiveness depends on the cellular environment [21]. Our results do not contradict these observations and identify both the human and mouse TRPA1 as molecular detectors of noxious heat and noxious cold. The bidirectional thermosensitivity of hTRPA1 was previously reported in [29]. Interestingly, the authors also noticed that in the presence of nonelectrophilic agonist carvacrol, the channels respond to cold only when they are pre-exposed to a warm (~37 °C) temperature. We have shown that the cold sensitivity of hTRPA1 and mTRPA1 can be triggered after the co-application of noxious heat together with depolarizing voltage. Thus, various sources of energy, such as that derived from agonist binding or depolarization, could drive the channel opening by cooling. The fact that the cold-sensitized channels are typically inhibited by heat suggests that their temperature sensors might be activated by heating while being inversely coupled to the channel opening.

In human TRPA1, a ‘gain-of-function’ variant (rs398123010) associated with pathological cold pain has been identified [18]. In vitro, this missense variant, hTRPA1-N855S, induced a ~5-fold increase in inward currents upon activation by cold and was accompanied by a negative shift of the voltage dependence of channel activation compared with wild-type hTRPA1. This residue is located in the S4–S5 linker, which is a critical domain that impacts the TRP-like helix and the S5 region around V875. Apparently, mutations within the S4–S5 linker and the lower cavity of the sensor domain have dramatic effects on the gating equilibrium and/or the response to voltage [34]. Conformational changes in this allosteric nexus may propagate to the TRP-like domain, affect the process of α–π transition in S6 [62], and ultimately lead to a reduction or increase in free energy required to shift the channel gate from the closed to the open state. Interestingly, the superposition of the TRPA1 (closed) structure with the recently published open-state structure of TRPV3 obtained at 37 °C [63] has shown a very good overlap within the S6 and the TRP-like domain. Thus, it is possible that the overall gating mechanism of thermosensitive TRP channels may be shared. During the review process, a structural study on TRPA1 was published identifying a direct interaction between N855 and the backbone carbonyl of C1024 from a short predicted C-terminal α-helix located at the cytosol-membrane interface near S1 and S4 of the sensor domain [64]. The findings support our previous predictions [56] and may partly explain our data presented in Appendix A.

Though more experimental support is required to unveil the exact molecular mechanism of the temperature-dependent activation of TRPA1, our data suggest the existence of a heat-sensitive domain in both human and mouse orthologues. The maximum slope of the temperature dependence of the responses induced by a heat-step protocol (Appendix A) was apparently higher than that obtained by heat ramps (Appendix A). Though the apparent *Q*_10_ reflects complex effects of temperature on channel gating kinetics and the applicability of such characteristics is limited [65,66], this observation is reminiscent to what has been recently described for the related channel TRPV3 [67]. If the rate of transition between the closed and open states of the channel is primarily driven by temperature, the direct measurement of the activation enthalpy would require a temperature stimulator that enabled the exchange of temperature at a very high speed (55 °C/ms). Most likely, the slower rate of the temperature exchange (~35 °C/s) compared to faster temperature steps (~110 °C/s) allowed the activated channels to transition to the state with a reduced energetics of gating. There is also no doubt that the speed of temperature exchange may be another source of variability among various reports (e.g., see [23]).

## Figures and Tables

**Figure 1 cells-09-00057-f001:**
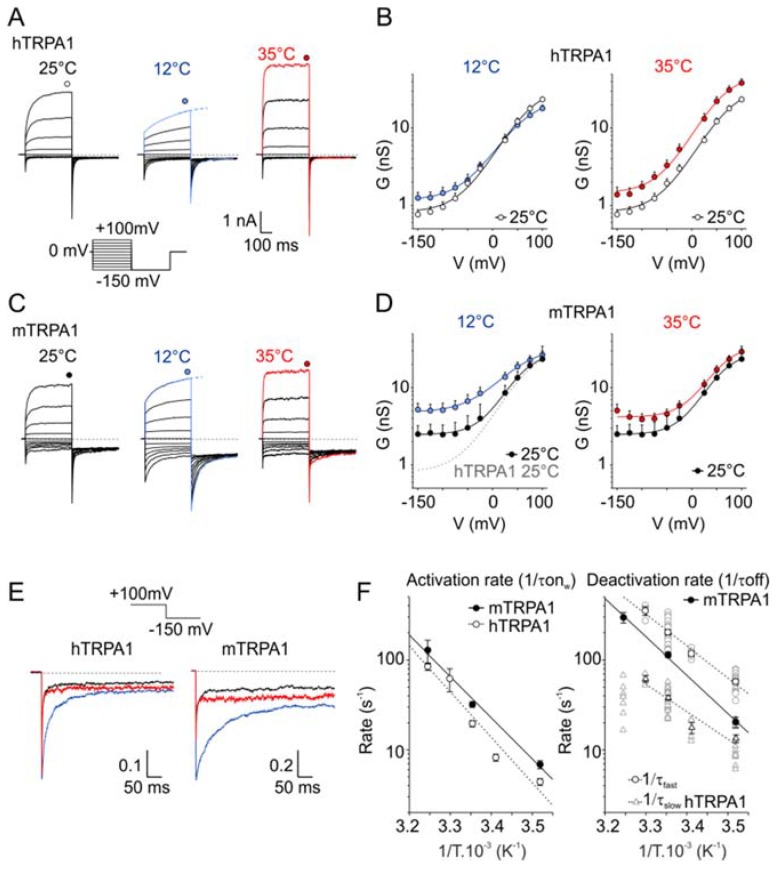
Effects of cooling and warming on the voltage-dependent gating kinetics of the human and mouse transient receptor potential (TRP) channel subtype A1 (TRPA1) orthologues. (**A**) Representative whole-cell currents in Ca^2+^-free intracellular and extracellular solutions recorded from a human embryonic kidney (HEK) 293T cell line transiently expressing the wild-type human TRPA1 (hTRPA1) channel, measured by the indicated voltage-step protocol at different temperatures. The voltage-step protocol was previously used by [17]. Steady-state currents were measured at the end of the pulses, as indicated by colored symbols atop each record. At 12 °C, the steady-state currents were estimated from the exponential fit of the recorded activation curves, as indicated by dashed lines. (**B**) The average conductance of hTRPA1 was obtained from the voltage step protocols, as in (**A**). The data represent the means + SEM. (*n* ≥ 6). The solid lines represent the best fit to a Boltzmann function, as described in Materials and Methods. (**C**) Representative whole-cell currents through wild-type mouse TRPA1 (mTRPA1), as in (**A**). (**D**) Average conductance of mTRPA1 obtained from voltage step protocols, as in (**C**). The data represent the means + SEM. (*n* ≥ 4). The solid lines represent the best fit to a Boltzmann function, as described in Materials and methods. The fit of hTRPA1 at 25 °C is overlaid as a grey dashed line for comparison. Note the increased basal conductance of mTRPA1 at negative potentials. (**E**) Comparison of the deactivation kinetics of TRPA1 orthologues at different temperatures. Representative tail currents normalized to the maximum amplitude at +100 mV obtained as an indicated part of the voltage step protocols, as in (**A**,**C**). Color-coded as the rest of the figure: black for 25 °C, red for 35 °C, and blue 12 °C. While the mTRPA1 tail currents could be fitted by a monoexponential function, the hTRPA1 tail currents could be well-fitted only by two exponentials. (**F**) Arrhenius plots of the onset (left) and deactivation (right) time constants for hTRPA1 (open symbols) and mTRPA1 (close symbols). The time constants of the onset were determined by mono- (mTRPA1) or bi-exponential (hTRPA1, plotted as weighted mean) fits to the time course of the onset currents at +100 mV, as in (**A**,**C**). The time constants of the deactivation rate were determined by mono- (mTRPA1) or bi-exponential (hTRPA1, fast and slow components plotted separately as open circles and triangles) fits to the tail currents, as in (**E**).

**Figure 2 cells-09-00057-f002:**
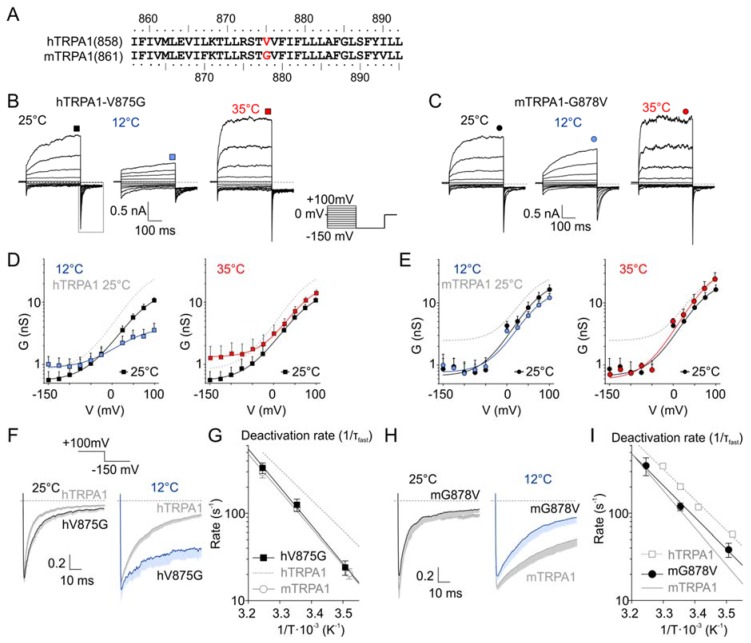
A single residue in S5 is involved in the species-specific differences in temperature sensitivity. (**A**) Sequence alignment of the fifth transmembrane domain (S5) of human and mouse TRPA1. The residues V875 (hTRPA1) and G878 (mTRPA1) underlying the species-specific differences in cold sensitivity are marked in red. (**B**) Representative whole-cell currents in Ca^2+^-free intracellular and extracellular solutions recorded from a human embryonic kidney (HEK) 293T cell line transiently expressing the human-to-mouse mutation hTRPA1-V875G, measured by the indicated voltage-step protocol at different temperatures. Steady-state currents were measured at the end of the pulses, as indicated by colored symbols atop each record. Dashed lines indicate zero current. (**C**) Representative whole-cell currents of mouse-to-human mutation mTRPA1-G878V, as in (**B**). (**D**) Average conductance of hTRPA1-V875G obtained from voltage step protocols, as in (**B**). The data represent the means +SEM. (*n* = 7). The solid lines represent the best fit to a Boltzmann function, as described in Materials and Methods. The fit of hTRPA1 at 25 °C is included as grey dashed line for comparison. (**E**) Average conductance of mTRPA1-G878V obtained from voltage step protocols, as in (**C**). The data represent the means + SEM. (*n* = 4). The solid lines represent the best fit to a Boltzmann function, as described in Materials and methods. The fit of mTRPA1 at 25 °C is included as grey dashed line for comparison. (**F**) Deactivation kinetics of hTRPA1-V875G compared with hTRPA1 acquired at 25 °C (left) and 12 °C (right). Average tail currents normalized to the maximum amplitude at +100 mV, obtained as an indicated part of the voltage-induced currents, as in (**B**); dashed box. The average currents are shown with gray or blue bars that indicate means –SEM. (*n* = 5). (**G**) Arrhenius plots of fast deactivation time constants of hTRPA1-V875G (black squares). The linear regression (black line) is steeper than that of hTRPA1 (gray dashed line) and resembles that of mTRPA1 (gray open circles and gray line). Error bars are SEM. (*n* = 5). (**H**) Deactivation kinetics of mTRPA1-G878V mutant compared with mTRPA1 acquired at 25 °C (left) and 12 °C (right), as in (**C**). Error bars are SEM. (*n* = 4). (**I**) Arrhenius plots of fast deactivation time constants of mTRPA1-G878V (black circles). The linear regression (black line) is less steep than that of mTRPA1 (gray line), and its steepness resembles that of hTRPA1 (grey squares and gray dashed line). Error bars are SEM (*n* = 4).

**Figure 3 cells-09-00057-f003:**
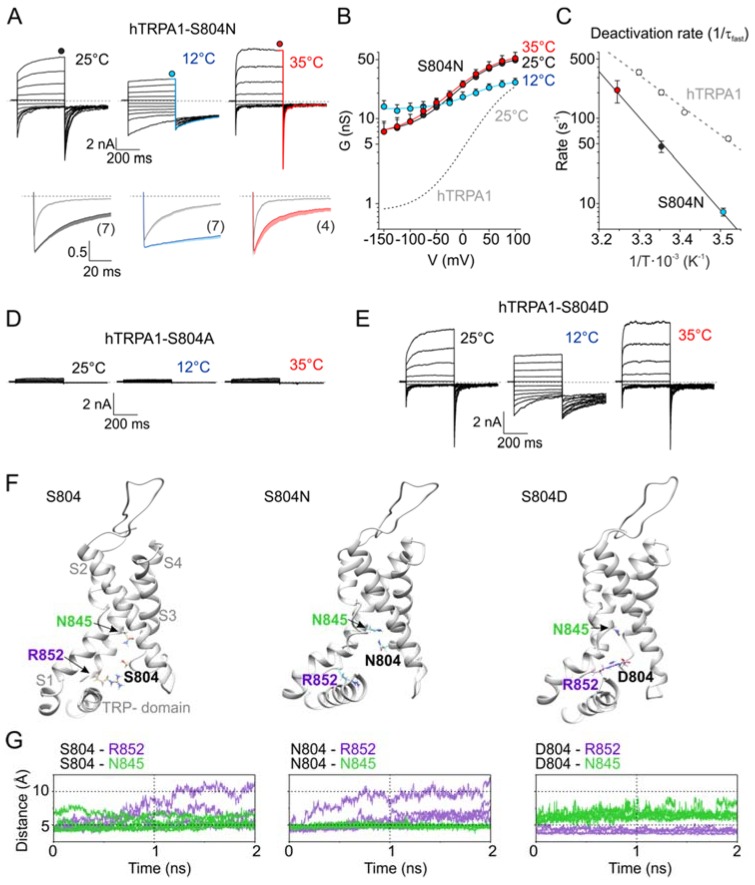
Functional screen for non-homologous residues in the vicinity of V875 identifies S804N as the most affected. (**A**) Representative whole-cell currents of mouse-to-human mutation mTRPA1-S804N obtained by voltage step protocol, as in Figure 1A. Below, the deactivation kinetics of S804N. Averaged tail currents normalized to the maximum amplitude at +100 mV (indicated by the circle above the record). The average currents are shown with color line and bars indicating means -SEM (number of cells indicated in parentheses). The gray lines with gray bars (-SEM) represent the averaged tail currents obtained from data for wild-type hTRPA1. Dashed lines indicate zero current. (**B**) Average conductance of hTRPA1-S804N obtained from voltage step protocols, as in (**A**). The data represent the means + SEM (*n* indicated in (**A**)). The solid lines represent the best fit to a Boltzmann function, as described in Materials and methods. The fit of hTRPA1 at 25 °C is included as grey dashed line for comparison. (**C**) Arrhenius plots of the fast deactivation time constants of hTRPA1-S804N (colored circles). The linear regression (black line) is steeper than hTRPA1 (gray dashed line). Error bars are ± SEM (*n* indicated in (**A**)). (**D**,**E**) Representative whole-cell currents of hTRPA1-S804A (**D**) or hTRPA1-S804D (**E**), obtained by voltage step protocol, as in (**A**). (**F**) Molecular dynamics (MD) simulation results obtained for wild-type TRPA1 (left), S804N (middle) and S804D (right). Snapshots from MD runs show the S1–S4 sensor domain and the TRP-like-box. (**G**) Time evolution of distances separating side chains at positions 804, 845 and 852 produced from MD simulations for all four subunits. Values lower than ~4 Å indicate the existence of inter-residue interactions (salt bridges and hydrogen bonds). Note that S804 alternates the interaction between R852 and N845. N804 prefers contact with N845, whereas D804 prefers contact with R852.

**Figure 4 cells-09-00057-f004:**
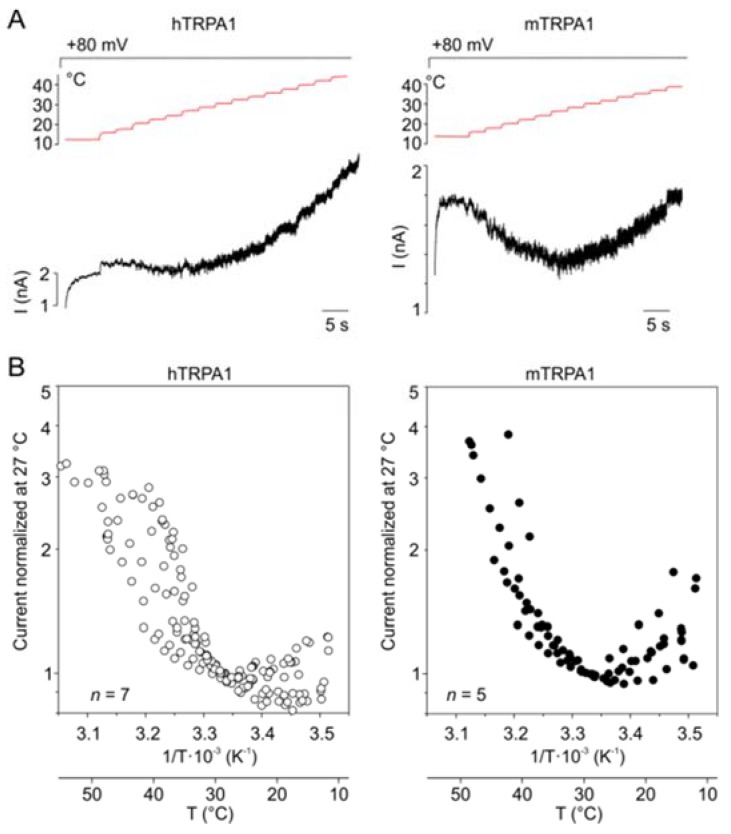
Both TRPA1 orthologues exhibited U-shaped outward currents. (**A**) Representative whole-cell currents of hTRPA1 (left) and mTRPA1 (right) elicited by the simultaneous activation of the voltage and temperature sensors at the holding potential of +80 mV and the temperature sensors by using increasing steps of temperature from ~11 to ~52 °C, as indicated above. (**B**) Arrhenius plots of currents normalized at 27 °C obtained from measurements, as in (**A**) for indicated (*n*) number of cells.

**Figure 5 cells-09-00057-f005:**
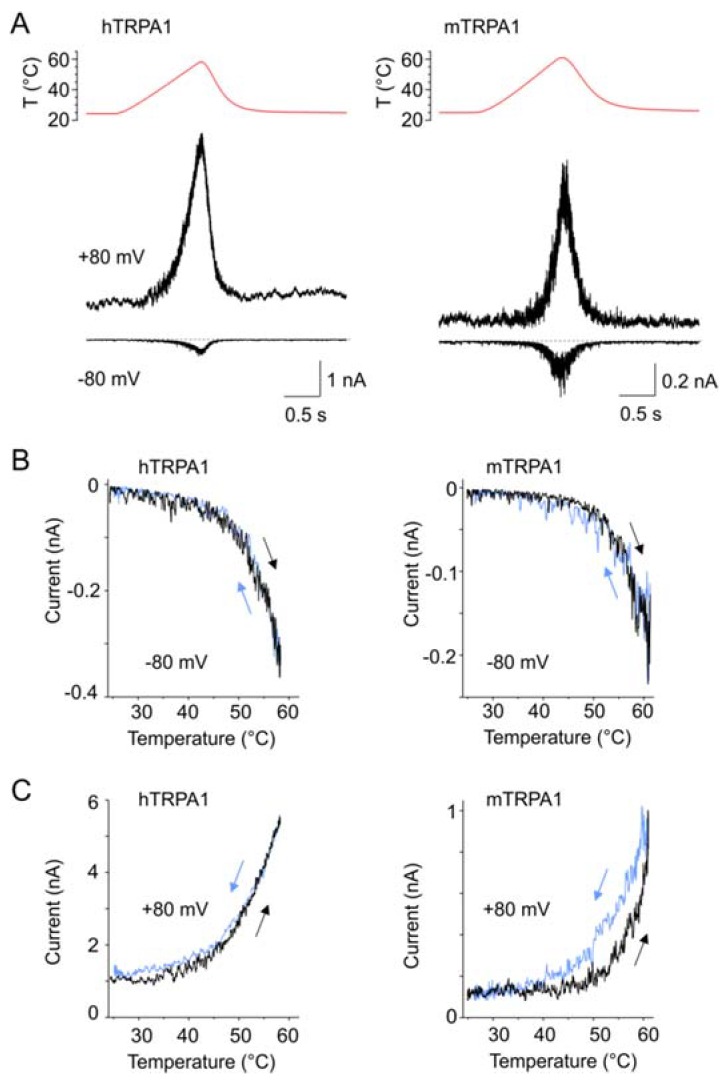
TRPA1 was directly and reversibly activated by heat at positive and negative membrane voltages. (**A**) Representative whole-cell currents of hTRPA1 (left) and mTRPA1 (right) activated by a temperature ramp (shown above the current traces) applied at a maximum speed of about 35 °C/s and at constant holding potentials +80 and −80 mV. (**B**,**C**) Currents measured at (**B**) −80 mV or (**C**) +80mV, as in (**A**), plotted as a function of temperature. The arrows indicate the direction of temperature increase (black) or decrease (blue). Note that the paths followed by activation and deactivation were almost identical, i.e., the activation–deactivation was accompanied by a low degree of hysteresis. Typical examples are shown for four (mTRPA1) and five (hTRPA1) similar recordings.

**Figure 6 cells-09-00057-f006:**
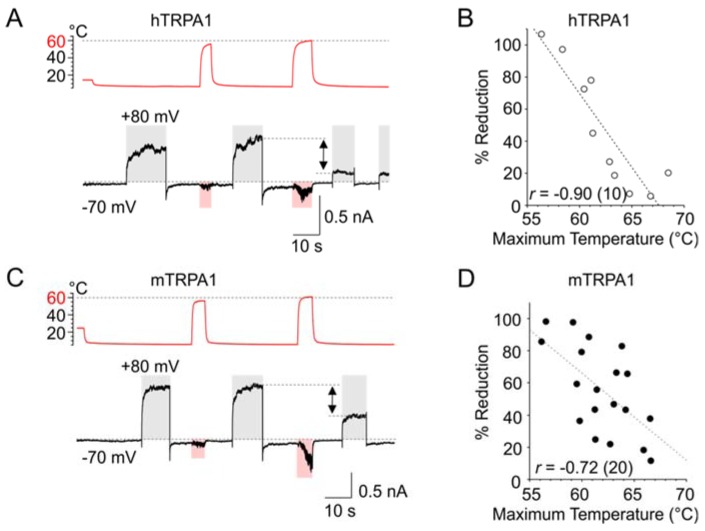
Temperatures above, but not below, 60 °C impede voltage-induced activation. (**A**) Representative whole-cell currents of hTRPA1 in response to temperature (shaded pink area) and voltage (shaded grey area) steps. The temperature trace is shown above the records. Repetitive depolarization from −70 to +80 mV elicited outward currents that were strongly reduced only if the heat step exceeded 60 °C. The time course of temperature changes (from 6 to 56 °C and then to 61 °C) is indicated above the record. The reduction is marked by arrow. (**B**) The degree of the reduction of voltage-induced currents was strongly correlated to the maximum temperature applied. (**C**) Representative whole-cell currents of mTRPA1 treated as in (**A**). (**D**) The current reduction of mTRPA1 was strongly correlated to the maximum temperature applied. The number of cells is indicated in brackets for each orthologue.

**Figure 7 cells-09-00057-f007:**
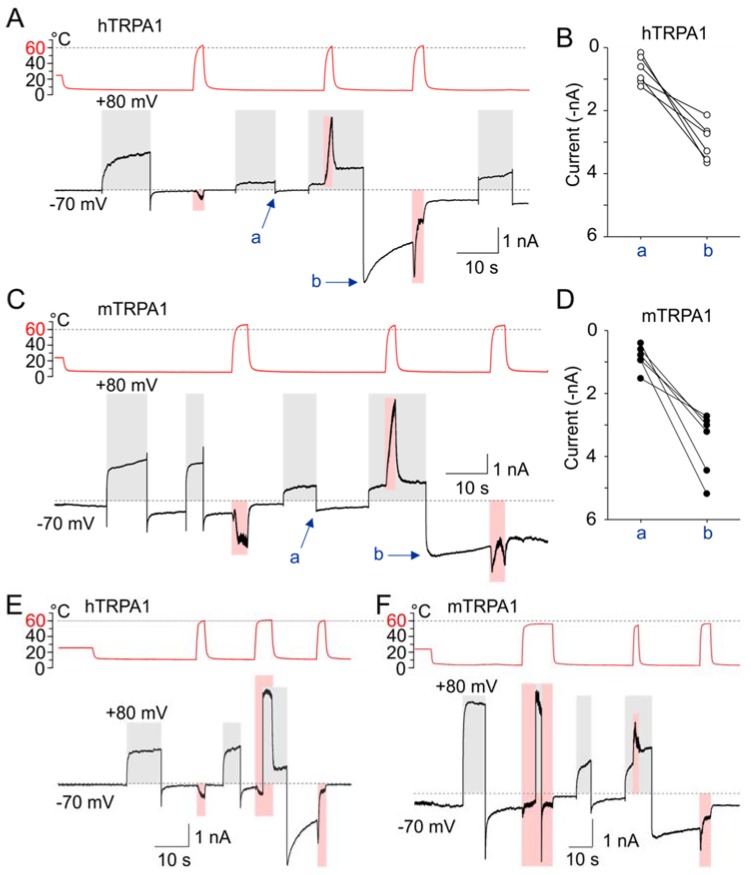
Concurrent activation of voltage- and heat-sensor renders the TRPA1 channel cold-sensitive. (**A**) Representative whole-cell currents of hTRPA1 in response to temperature (shaded pink area) and voltage (shaded grey area) steps. The temperature trace is shown above the records. The current-to-temperature relationship of the first heat response is shown in Appendix A. Exposure to depolarization concurrently with excessive heat rendered the TRPA1 channels dramatically cold-activated (arrow b) once back at negative membrane potentials, as compared to inward current (arrow a) after exposure to only depolarization. (**B**) The increase in inward currents for individual cells expressing hTRPA1 (*n* = 6), measured at times indicated by arrows in (**A**). (**C**) Representative whole-cell currents of mTRPA1 treated like those in (**A**). (**D**) The increase in currents for six individual cells expressing mTRPA1 measured at times indicated by the arrows in (**C**). (**E**) Representative whole-cell currents of hTRPA1 treated similarly to those in (**A**). The cold-induced inward currents were produced independently of whether depolarization or heat was applied prior to the concurrent activation of both sensors. (**F**) Representative whole-cell currents of mTRPA1 treated similarly to those in (**A**). Note the difference in inward current after depolarization during noxious heat exposure or after noxious heat exposure during depolarization.

**Figure 8 cells-09-00057-f008:**
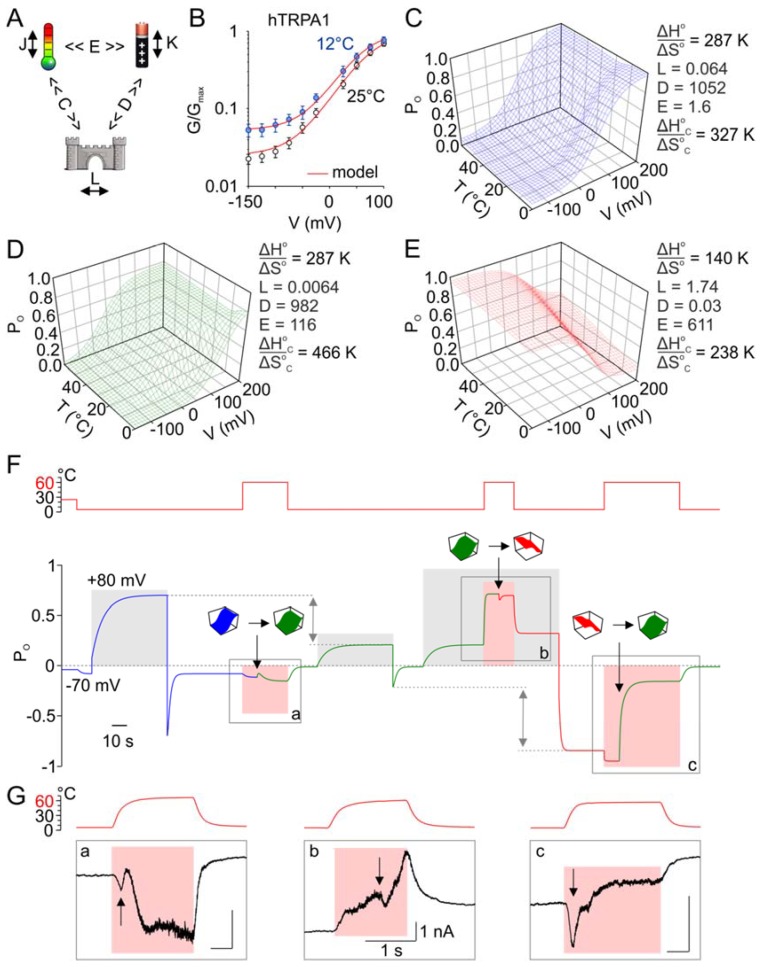
The eight-state allosteric model is adequate for explaining the variability in TRPA1 temperature responsiveness. (**A**) Allosteric model for temperature- and voltage-dependent activation of TRPA1. The transitions between open and closed states are given by the intrinsic equilibrium constant *L*. The transitions between resting and active conformations of temperature sensors and voltage sensors are given by equilibrium constants *J*(*T*) = exp[−(Δ*H* − *T*Δ*S*)/*RT*] and *K* = *K*(0)exp(*zFV*/*RT*), respectively. The coupling of the voltage sensor to the channel gate and to the temperature sensor is governed by the allosteric coupling constants *D* and *E*, which are assumed to be temperature-independent. The coupling of temperature sensor to the channel opening is assumed to be temperature-dependent by the term *C*(*T*) = exp[−(Δ*H*^o^*_C_* − *T*Δ*S*^o^*_C_*)/*RT*]. (**B**) Normalized conductance of hTRPA1 (same as in Figure 1B) at the indicated temperatures. Red lines are the global fit by using the model in (**A)**. Parameters of the fit were used to plot open probability landscape in (**C**); for details, see Materials and Methods. (**C**) Open probability landscape obtained by calculating *P*_o_ as a function of temperature and voltage by using the parameters from fit in (**B)**: *K*(0) = 0.003, *z* = −0.74, Δ*H*^o^_C_ = 6.1 kcal mol^−1^ and value Δ*H* = 91 kcal mol^−1^ published for TRPM8 [39]. Note a mild U-shape thermosensitivity. (**D**) ‘Irreversible switch-after-excessive heat‘ mechanism. Exposure to heat over 60 °C triggers the reduction of currents at +80 mV (as in Figure 6A,C). Open probability landscape was remodeled to qualitatively match the activation profile of hTRPA1, as shown in Figure 7A. The parameters *K*(0)*, z* and Δ*H*^o^ were preserved from blue *P*_o_ landscape in (**C**), Δ*H*^o^*_C_* = 1.4 kcal mol^−1^. (**E**) Concurrent activation of voltage- and heat-sensor rendered TRPA1 sensitized to cold. The open probability landscape obtained in (**D)** was remodeled to approximate the experimentally observed data. The parameters *K*(0), *z* and Δ*H*^o^ were preserved from blue *P*_o_ landscape in (**C**), Δ*H*^o^*_C_* = 4.3 kcal mol^−1^. (**F**) The model is capable of capturing all essential aspects of our observations. The time course of *P*_o_ modelled according to Figure 7A. The blue, green and red parts of the *P*_o_ trace were modelled by using the respective parameters from (**B**,**D**,**E**). The transitions between landscapes are indicated by black arrows and colored pictograms. The depolarizations to +80 mV are marked as shaded grey areas, and the applied temperature is shown above as red trace and shaded pink areas for 60 °C. Grey bidirectional arrows show the reduction of *P*_o_ (as in Figure 6A,C) after the first exposure to heat at −70 mV, and the increase in *P*_o_ (as in Figure 7A) after the second exposure to heat at +80 mV. Grey rectangles marked as a, b and c, correspond to panels in (**G**). (**G**) Representative whole-cell currents of mTRPA1 in response to noxious heat temperature. The situations correspond to those in (**F**), and the holding potential was −70 mV in a and c and +80 mV in b. Black arrows point to current transients that were captured also by our model in (**F**). The time course of temperature is indicated above the records. The bars indicate 1 s and 1 nA in panels a, b, and c.

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
