# Peer review of "Human and Mouse TRPA1 Are Heat and Cold Sensors Differentially Tuned by Voltage"

_cells, 2019, doi:10.3390/cells9010057_

Round 1

Reviewer 1 Report

The authors investigate in their manuscript the activation of the TRPA1 ion channel by high and low temperatures. The TRPA1 channels is an important sensor for several noxious stimuli, including heat and cold, in peripheral sensory neurons and therefore plays an key role in the nociceptive system. The study presented in this manuscript aims to unravel the molecular mechanisms underlying the complex activation of this ion channels.

The study is very well made and is methodically and technically sound. The results are highly valuable, since they help to explain the polymodal activation system of TRPA1. The discussion section explains the findings nicely in the context of previously published data.

The major problem of this study is that it has been conducted solely using TRPA1 overexpressed in a heterologous system. Although some of the experiments need to be performed in this setting, i.e. the testing of TRPA1 mutations, it should be kept in mind that the findings with ion channels in heterologous systems are not always translating to primary cells.

Thus, I would strongly recommend to repeat selected key experiments using primary murine DRG neurons. In particular the concurrent activation of voltage- and heat-sensor causing TRPA1 cold sensitization would be of interest.

Author Response

Referee: The major problem of this study is that it has been conducted solely using TRPA1 overexpressed in a heterologous system. Although some of the experiments need to be performed in this setting, i.e. the testing of TRPA1 mutations, it should be kept in mind that the findings with ion channels in heterologous systems are not always translating to primary cells.

Thus, I would strongly recommend to repeat selected key experiments using primary murine DRG neurons. In particular the concurrent activation of voltage- and heat-sensor causing TRPA1 cold sensitization would be of interest..

-- Response: We would like to thank Reviewer #1 for carefully reading our manuscript and for a very useful and constructive comment. Below are our responses and a detailed description of the changes made to the manuscript.

We have conducted additional electrophysiological experiments as recommended. We measured currents from F11 cells transfected with TRPA1 and found that the key findings observed with HEK293T cells were also seen with F11. This cell line is a suitable model which maintains broad functional similarities with dorsal root ganglia neurons (1, 2). We reasoned that using primary sensory (dorsal root ganglia) neurons would lead to unambiguous interpretation of the results because of the presence of other temperature sensitive channels (e.g. see (3, 4)). To test the physiological relevance at neuronal level would require more extensive experimental approach (e.g. use of specific channel agonists and antagonists, reporter mice, etc…) which, to our opinion, would exceed the scope of our study and would deserve its own report.

In the revised version of the manuscript, we have added a new chapter 3.9 and a supplementary Figure S7, in which we describe new results on F11 cells. We now state (lines 609-630):

„3.9 Activation of TRPA1 in F11 cells

To investigate whether TRPA1 can be activated by heat and cold under close-to-native conditions, we used F11 neuroblastoma cells derived from dorsal root ganglia neurons which provide a well characterized cellular model of peripheral sensory neurons [58]. We transfected the cells with mouse or human TRPA1 and measured currents at -70 mV using a similar temperature and voltage protocol as shown in Figure 7A. In addition, the TRPA1 inhibitor HC-030031 (50 µM) was used to assess the extent of specific contribution of TRPA1 to temperature-dependent currents. We observed very similar current characteristics as those seen in channels expressed in HEK293T cells. Although the application of heat above 60 °C was experimentally challenging and inevitably destroyed the seal in many cells (19 of 29), we succeeded in recording the typical responses to heat and voltage in 8 neurons expressing mTRPA1 and 2 neurons expressing hTRPA1 (Figure S7). In both TRPA1 orthologues, we observed a reduction in currents induced by depolarizing step to +80 mV only when the preceding heat step exceeded 60 °C. The responses activated during heating typically exhibited current transients, suggesting that the channels pass through several conformational states as in HEK293T cells (see Figure 8G). When the neurons were depolarized and concurrently stimulated by noxious heat, inward currents arising from cooling the cell to 5-7 °C and repolarizing it to  70 mV were increased (Figure S7B) and slowed down (Figure S7A,C).“

We thank the Reviewer for this suggestion and agree that such experiments substantially strengthen our study.

References:

Vetter, I., and Lewis, R. J. (2010) Characterization of endogenous calcium responses in neuronal cell lines, Biochem. Pharmacol. 79, 908-920. Yin, K., Baillie, G. J., and Vetter, I. (2016) Neuronal cell lines as model dorsal root ganglion neurons: A transcriptomic comparison, Mol. Pain 12. Weng, H. J., Patel, K. N., Jeske, N. A., Bierbower, S. M., Zou, W., Tiwari, V., Zheng, Q., Tang, Z., Mo, G. C., Wang, Y., Geng, Y., Zhang, J., Guan, Y., Akopian, A. N., and Dong, X. (2015) Tmem100 Is a Regulator of TRPA1-TRPV1 Complex and Contributes to Persistent Pain, Neuron 85, 833-846. Akopian, A. N. (2011) Regulation of nociceptive transmission at the periphery via TRPA1-TRPV1 interactions, Curr Pharm Biotechnol 12, 89-94.

Reviewer 2 Report

Transient receptor potential channel subtype A1 (TRPA1) is involved in thermal and chemical nociception. Originally considered a temperature-insensitive channel, TRPA1 has turned out to be capapble of sensing both, cold and heat.

Within their work, and by the use of electrophysiology and molecular dynamics simulations, Sinica et al compare the properties of human and mouse TRPA1 aiming at deciphering fundamental differences between gating properties of the 2 proteins. They show that both channels are activated by heat and propose an allosteric mechanism that accounts for the variability in TRPA1 temperature responsiveness.

All experiments shown are carried out with utmost care and do not lack any controls. The paper is clearly written and results, especially the heat-induced cold sensitivity of TRPA1, are novel and deliver important new knowledge to the TRP-field.

I only have some minor concerns to improve the quality of the manuscript

Figure 2: it should be mentioned in the text (starting at line 288) that the hTRPA1-V875G mutation is the corresponding one to mTRPA1-G878V. In addition, I recommend to add a short sequence alignment of rodent and primate TRPA1 to the figure (or maybe move Suppl. Fig. S2 to main text). For the S804 substitutions tested – have there been more than N, A and D tested? To propose side chain size and polarity determining the gating kinetics would require more tested residues differing in size and charge. By the use of MD-simulations the authors identify loose contacts of S804 to R852 and N845 in wildtype-channels – have the authors tested if channel gating is altered upon mutation of those residues?

Author Response

We would like to thank the Reviewer for her/his time and for very helpful, constructive comments. Please find below our responses to each of the comments, as well as a detailed description of the changes made to the manuscript.

Referee: Figure 2: it should be mentioned in the text (starting at line 288) that the hTRPA1-V875G mutation is the corresponding one to mTRPA1-G878V. In addition, I recommend to add a short sequence alignment of rodent and primate TRPA1 to the figure (or maybe move Suppl. Fig. S2 to main text).

-- Response: We have added a sequence alignment of the fifth transmembrane domain of human and mouse TRPA1 to Figure 2A. The text starting at line 288 now refers to the Figure 2A and clarifies this issue: “..Previous studies have identified G878 in mTRPA1 and V875 in hTRPA1 (Figure 2A) as residues underlying the species-specific differences in cold sensitivity [26].”

Referee: For the S804 substitutions tested – have there been more than N, A and D tested? To propose side chain size and polarity determining the gating kinetics would require more tested residues differing in size and charge.

-- Response: We have tested only the mutants S804D, S804A and S804N. We agree with the argument and have omitted the critical sentence (line 348).

Referee: By the use of MD-simulations the authors identify loose contacts of S804 to R852 and N845 in wildtype-channels – have the authors tested if channel gating is altered upon mutation of those residues?

-- Response: In fact, we have tested mutations at both these residues. We have previously demonstrated [Ref. 34] that mutations at R852 critically affect the gating of human TRPA1. Mutation R852A rendered the channels less sensitive to voltage and chemical stimuli. On the other hand, the R852E mutation produced channels that we called a “separation-of-function phenotype“. This construct exhibited an increased basal channel activity, a loss of calcium-induced potentiation and an accelerated calcium-dependent inactivation. We have also demonstrated that the functionality of R852E can be affected by the conservative mutation at K868R, which we have interpreted as that these two residues are on the same allosteric pathway.

In the frame of the current study, we have tested also the N845I mutant. This substitution was prepared because we originally aimed to test the hypothesis that increasing the hydrophobicity within the S1-S4 intracellular cavity may affect the temperature sensitivity of TRPA1. This general hypothesis was based on our own observations [see Ref. 42] and on the “dynamic hydration” mechanism proposed previously by several authors e.g. Clapham and Miller (2011), Proc. Natl. Acad. Sci. USA. 108, 19492-19497 and Kasimova et al. J Gen Physiol (2018) 150 (11): 1554-1566. The N845I mutant, however, rendered the channel much less responsive to all stimuli, which supported the hypothesis. However, the cold-induced effects could not be reliably evaluated particularly at negative membrane potentials where the currents are very small and interfere with endogenous currents from HEK293T cells (please see the enclosed N845I.pdf demonstrating the conductance of a representative cell expressing N845I). Therefore, we decided not to include these results into the study.

Our previously published results suggest that the lower vestibule of the S1–S4 sensor domain is an important component of an allosteric nexus at which various activation signals are integrated and transmitted through the TRP-like domain to the intracellular channel gate [Ref. 42]. Structural comparisons indicate that isoleucine at position 845 may together with Leu807 and Ile811 form a rigid hydrophobic cluster that affects the energy transfer between sensing domains and channel pore, thereby hindering the gating process. A similar functional deficit produced by mutation R852A could be due to hydrophobic interactions between alanine at position 852 and Leu982 from the TRP-like domain. We are fully aware that further experiments starting with appropriate mutagenesis studies are worthy of future investigation.

Round 2

Reviewer 1 Report

I do not have further comments